# Determining the influence of LPI, GCI and IR on FDI: A study on the Asia and Pacific Region

**Pasindu Wannisinghe**[1], **Sanjula Jayakody**[1], **Sashini Rathnayake**[1], **Deshani Wijayasinghe**[1], **Ruwan Jayathilaka**[2]*, **Naduni Madhavika**[2]

**1** SLIIT Business School, Sri Lanka Institute of Information Technology, Malabe, Sri Lanka, **2** Department of Information Management, SLIIT Business School, Sri Lanka Institute of Information Technology, Malabe, Sri Lanka

* ruwan.j@sliit.lk

**Data Availability Statement:** All relevant data are within the paper and its with Supporting information files.

## Abstract

This study examines the impact of the Logistics Performance Index (LPI), Global Competitiveness Index (GCI) and Interest Rates (IR) on Foreign Direct Investment (FDI) for the Asia & Pacific region. The study is original as extensive evidence on the impact of LPI, GCI and IR on FDI in the Asia & Pacific region are examined initially. For the years 2007, 2010, 2012, 2014, 2016 and 2018, data was gathered for 33 nations in the Asia and Pacific area. Data analysis was performed using a panel regression model and multiple linear regression. The findings of the study reveal that LPI, GCI and IR are the three major factors influencing FDI inflows into the economies. However, the impact of these factors varies from country to country. The results concluded that LPI positively impacts FDI in India, Korea, Lebanon, and Oman. In contrast, a negative influence was observed for China, Kuwait and the Philippines. GCI positively impacts FDI in China, Korea, Kuwait, Pakistan and the Philippines, while a negative impact was observed in Armenia, India, Lebanon. Furthermore, IR has a positive impact on FDI flows in China and Egypt while in Korea and Lebanon, a negative impact was observed. Therefore, policymakers should focus more on improving the infrastructural requirements and macroeconomic factors while considering the other country-level variables that influence the FDI in flow.

## Introduction

The rapid growth of global Foreign Direct Investment (FDI) flow initiated since the early 1980s and currently, it has become one of the most crucial modes of attracting external capital towards economies from the private sector [1]. In general, FDI is a form of cross border investment where investors residing in one country invest in another country/economy and can significantly influence investee entity's economic activities entity beyond the border [2]. In recent years, global FDI has become major phenomenon owing to economic interconnectivity, globalisation, the rapid growth of international trade, inter-regional and intra-regional trade agreements, among others [3]. As per the UNCTAD [4], developing countries have attracted a significant portion of the FDI inflow. Likewise, Buthe and Milner [5] specified that the reason

**Funding:** The authors received no specific funding for this work.

**Competing interests:** The authors have declared that no competing interests exist.

developing countries attract a greater portion of global FDI is high return on investment and convenience of conducting business due to investor-friendly economic conditions, such as trade and finance openness and low level of government intervention.

However, during the last five years, the FDI flow has been reducing drastically whereas in 2020, it dropped drastically by 56%. However, the FDI flows in developing Asia have decreased only by 6%, proving it to be the ideal FDI destination [6]. Therefore, Asia can be defined as one of the major FDI destinations. Abundant studies conducted have analysed the determinants of FDI in the Asian region and this study suggests that logistic and infrastructural performance, the competitiveness of countries and Interest Rates (IR) have a considerable impact on Asia and Pacific regional FDI flow.

The Logistic Performance Index (LPI) global benchmarking tool created and managed by the World Bank is one of the most sophisticated tools that enable comparison of the trade logistics performance and infrastructural quality of countries at an extensive level under seven different indicators [7]. The indicators include the overall value, quality of custom services, infrastructure, international shipments, quality of logistics, ability to track and trace and lastly, the timeliness of the services. Moreover, LPI is a vital index to evaluate countries' competitive position for developing countries that are seeking to enhance their infrastructural developments to accelerate the country's growth in order to facilitate a high level of international integration [8]. The global supply chain is a major phenomenon in the modern-day business world. On the one hand, Martí, Puertas [9] infer that LPI focuses on all the crucial factors in the logistic sector and has contributed to identifying policy reforms, integrated development measures and many more enhancements to facilitate streamlined logistic services, considering both domestic and international perspectives. On the other hand, logistic capabilities and transportation have become major determinants of attracting FDI and achieving sustainable economic growth [10].

The Global Competitiveness Index (GCI) is an extensive indicator that facilitates countries to evaluate their productivity and growth from short and long-term perspectives. In doing so, it focuses on microeconomic and macroeconomic factors which drive countries towards prosperity [11]. The GCI index for over 130 countries was analysed and presented under 12 dissimilar competitiveness pillars, including institution, infrastructure, ICT adoption etc. Lysandrou, Solomon [12] emphasise that GCI is widely accepted in research to determine public sector quality as well as private sector competitiveness positions.

Apart from these, IR can be considered one of the critical factors influencing the investment flow [13]. Binsbergen (2022) stresses that IR is a recognised tool to calculate the time value of money or to determine the investment yield from a futuristic perspective. Diverse interest rates are considered in the international financial system, including lending IR, nominal IR, borrowing IR, and Real Interest Rate (RIR). According to Singhania and Gupta [14], the effect of IR on investors' investment behaviour, could vary based on the investment type the investors are focused on. If the investors are seeking finances from the internal finance market, those type of investors may focus on borrowing IR, while those seeking to inject money into another financial market would be concerned more on borrowing IR. As per the UNESCWA [15], RIR can be considered the most suitable form of IR for this analysis. This is because IR is calculated by eliminating the variations that could occur due to inflation to precisely determine the cost incurred by the borrower or the gain incurred by the fund lender.

Hence, LPI and GCI are justified as two globally recognised benchmarking tools. Here, LPI provides comprehensive insights on trade logistic performance and competitiveness of countries for governments and global enterprises to strategise their investment decisions; IR is a key component of a country's monetary policy that impact investment decisions. Therefore, it raises the question of how these factors individually and collectively influence the FDI flow in

the Asia and Pacific region. To further investigate the impact, this study objectified determining the impact of LPI, GCI and IR on FDI flow in the Asia and Pacific region. Consequently, the current paper seeks to contribute the following to the existing literature. Firstly, the study analyses the overall impact of LPI, GCI and IR on FDI, focusing on Asia and Pacific region countries, where a lacuna in the literature was observed. Secondly, the study provides a country specific analysis focusing on Asia and Pacific countries in determining how these three independent variables influence the investment flow within countries- i.e. how these three major factors determine the quality and the productivity of macroeconomic and microeconomic conditions within countries. Thirdly, the study focuses on understanding the differences in the linear trends of how all three variables impact countries individually, utilising graphical illustrations to visualise the trends.

The article's structure includes the following sections: an introduction to the topic and variables, a holistic literature review on the underlying variables concerning the Asia and the Pacific region, the data and methodology, the results of the analysis, a comprehensive discussion of findings, and the conclusion and recommendations.

## Literature review

Regarding the variables that determine foreign investment, the current literature emphasises the importance of several macroeconomic factors and other elements at the national or regional level that draw in FDI. Reviewing previous literature on the effects of LPI, GCI and IR on FDI, we conducted a systematic literature review for the Asia & Pacific area.

During the last few decades, the Asia and Pacific region has been attracting a significant portion of global FDI and as per UNCTAD [4], the Asia region has absorbed USD 619 billion. Moreover, LPI is considered a factor to positively impact FDI inflows into a country since adequate infrastructure facilities, efficient transportation systems, etc., boost a country's logistics performance. Consequently, trade and investments will thrive since better logistics performance facilitates trade between countries. Also, investors prefer to invest in countries with better infrastructure and logistics facilities, considering these lucrative. Continuous infrastructural developments tend to create an investor-friendly environment, as justified by Soh, Wong [16], where the findings identified a significant positive impact of LPI on FDI by employing the random effect model. The authors further highlighted that the Asian region's investment behaviour might not be influenced by LPI alone. In addition to that, the author disclosed that when considering the theoretical perspective of FDI, theories such as Dunning's eclectic paradigm theory which emphasise the fluctuations of FDI inflow with relation to multiple factors including efficiency which significantly rely on competitiveness, infrastructural capabilities, logistic performance and quality of governance is key theory of FDI aligning with study. Similarly, An, Razzaq [17] and Saidi, Mani [10] concluded that logistics performance is a determinant of FDI. An Indian study revealed that to increase FDI inflow and export development, Indian officials have focused more on expanding all types of logistical infrastructure [18]. Affirming the above-explained findings, Avioutskii and Tensaout [19] examined the importance of logistics infrastructure in attracting FDI into the European region in a similar study. Similarly, Souza, Goh [20], highlighted that multinational firms are one of the many parties involved in FDI, contributing significantly to global FDI flows. For these corporations, choosing where to invest has been heavily influenced by the host country's logistic performance. Despite the other factors, better administration, well-organised transportation networks, infrastructure, and logistical performance enhance international commerce, and FDI inflows that guarantee reducing unnecessary costs and uncertainties connected with both inter-regional and intra-regional financial activities, positively affect FDI growth [8, 21, 22].

The United Arab Emirates (UAE) in the Middle East and Vietnam in Asia are recognised as attracting FDI largely due to their infrastructure excellence and LPI [23, 24]. Furthermore, Shah [25] had identified that infrastructure availability in the host country has a positive impact on the location choices of foreign investors. Further, the author determined that macroeconomic management and economic development positively impact FDI, while high inflation affects FDI negatively. Also, according to Shah [26] better infrastructure and trade liberalisation encourage more FDI inflows into a country. These studies have emphasised how important LPI is in influencing FDI in the Asia-Pacific area.

Similar to LPI, GCI too positively impacts FDI inflows since factors such as institutional quality, infrastructure, macroeconomic stability, skills, innovation etc., of a country tend to encourage investors to invest in such countries [27]. A few research has taken place in the Asia & Pacific region To determine the influence of GCI on FDI. Due to China's quick rise in competitiveness, maintaining Thailand's level of competitiveness is essential for Thai businesses. The inability to reach international standards has been a major problem in recent years. Both the GCI and the Current Competitiveness Index (CCI) have been dropping over time, along with FDI inflow [28]. However, multiple authors point out that FDI is a substantial predictor of GCI, suggesting a potential bidirectional relationship [29, 30]. Additionally, numerous studies conducted in other regions infer that GCI has a considerable impact on FDI inflow as well [27, 31, 32]. Moreover, GCI includes many other variables under its sub pillars. When considering the impact of those variables on FDI, trade and investment liberalisation, market size, development level, human capital, political stability, regulatory quality, the openness of the host economy and good governance have a positive impact on FDI while corruption negatively impacts FDI [33, 34]. Therefore, the competitiveness position is a core area to improve in attracting investors.

As a crucial macroeconomic element, IR significantly impacts how investors invest and how much money they move worldwide. The impact of IR on FDI depends on the type of IR under consideration. Borrowing IR negatively impacts FDI inflows since investors tend to invest in countries with low borrowing IR to lower the cost of capital. In contrast, lending IR positively impacts FDI since investors tend to invest in countries with higher IR to ensure a higher return on their investments [35]. A study conducted in Australia established that IR has no significant impact on FDI [36]. In contrast, Metwally [37], found that IR has a significant impact on FDI in Jordan, Oman and Egypt, suggesting that the authorities should focus more on stabilising IR. Apart from this, multiple studies have been carried out in South Asian countries and according to Dhannur and John [38], a study in India shows that IR has a significant positive impact on foreign investment. In contrast, a similar study identified that IR negatively influences FDI in India [39, 40]. However, another study in India proved IR has an insignificant relationship [14]. A Bangladesh study shows that IR significantly positively affects FDI [41]. However, a similar study in Bangladesh shows IR significantly negatively impacts foreign investments [42]. According to Wijeweera and Mounter [43], FDI in Sri Lanka has been significantly and positively impacted by IR. Recent research indicated that while an increase in IR increases FDI influx to Southeast Asia, it also negatively impacts other macroeconomic indicators [44, 45]. However, Siddiqui and Aumeboonsuke [46] examined the long-term impact of IR on FDI inflow in Southeast Asian nations over a 25-year period, from 1986 to 2012, which revealed a considerable adverse impact on FDI. According to a study, appealing IR by itself will not substantially affect FDI flow in countries with heavy capital controls, including China, a nation recognised for having strict capital controls. This finding follows the fact that different outcomes are seen within the same area [47]. Moreover, Mishra and Jena [13] and Chandra and Handoyo [48] considered the entire Asian region, and the IR indicated an insignificant impact on FDI inflows. In addition, economic development and IR risk positively impact

Foreign Portfolio Investments (FPI), while inflation, exchange rates and country risk have a negative impact on FPI [49]. The variations like relationships between nations and regions in Asia and the Pacific demonstrate how the influence of IR varies depending on other macroeconomic factors in each country. Furthermore, multiple theories related to cross border investment aligning with macro-economic factors have also been covered under the scope of this study. Suhendra, Istikomah [50] emphasised the internal fund theory, according to which, the investment decision is based on the level of returns considering multiple factors and the return on investment including IR. On the other hand, Ajija and Fanani [51] pointed out that the Keynes theory infers that when the IR increases, the FDI inflow reduces while indicating a negative impact.

Past literature stresses that limited studies have analysed the individual impact of LPI, GCI and IR on FDI in the Asia & Pacific region. Moreover, according to our knowledge, no studies were conducted to analyse the overall impact of all three variables on FDI inflow. Therefore, the current study is dedicated to filling the above-mentioned research gap in the existing literature through a comprehensive analysis by determining the impact of LPI, GCI and IR on FDI in the Asia & Pacific region.

## Data and methodology

This study was reviewed and approved by Sri Lanka Institute of Information Technology (SLIIT) Business School and the SLIIT ethical review board. Study used the secondary data sources and the data file used for the study is presented in S1 Appendix. The research used a panel dataset of a six-year period from 2007 to 2018, focusing on 33 Asian & Pacific region countries. The period includes the years 2007, 2010, 2012, 2014, 2016, and 2018 with uneven gaps between years since the LPI data were published only for the above-mentioned six years. The inclusion of the countries can be specified as Asian countries, Middle Eastern nations and Australia and New Zealand, concentrating in the Pacific region. The approach utilised to determine the number of countries strongly relied on the availability of secondary data for variables. Furthermore, secondary data employed in the study were obtained from the World Development Indicators for LPL FDI and IR published by the World Bank and for GCI, the Global Competitiveness Report published by the World Economic Forum. Here, LPI includes 6 sub pillars such as customs, infrastructure, international shipments, quality of logistics services, tracking and tracing and timeliness, while GCI includes 12 major pillars, such as institutions, infrastructure, ICT adoption, macroeconomic stability, health, skills, product market, labour market, financial system, market size, business dynamism and innovation capability. Under these 12 pillars, there are another 98 sub pillars [52, 53]. Eq 1 will be employed to determine the impact of LPI, GCI, IR on FDI for the Asia & Pacific region and Eq 2 was formed to analyse the country-specific impact LPI, GCI, IR on FDI on FDI.

$$lnFDI_{it} = \beta_0 + \beta_1 LPI_{it} + \beta_2 GCI_{it} + \beta_3 IR_{it} + \varepsilon_{it} \tag{1}$$

$$lnFDI_t = \beta_0 + \beta_1 LPI_t + \beta_2 GCI_t + \beta_3 IR_t + \varepsilon_t \tag{2}$$

The *lnFDIit* denotes the natural log value of the FDI inflow. To ensure the constancy of data and to verify the data is normally distributed, the natural log values were utilised. Furthermore, FDI was measured by the current USD. *LPIit* indicates the overall LPI value, *GCIit* represents the overall GCI value, and *IRit* represents the RIR. The *i* represents the country in the panel data analysis, and the *t* denotes the time period. The coefficients of LPI, GCI and IR are presented by $\beta_1$, $\beta_2$ and $\beta_3$, respectively, while $\beta_0$ denotes the intercept. The $\varepsilon it$ indicates the error term of the regression equation.

The analytical technique applied in the research is regression analysis, and for the regional analysis, a panel data del was employed, while for the country-specific analysis, a Multiple Linear Regression (MLR) was utilised. As per Jayathilaka, Jayawardhana [54], the limited number of time series is a constraint in utilising non-stationary and dynamic panel data models. Therefore, it was determined that the most suitable panel regression models are Pooled Ordinary Least Square (POLS), Random Effect (RE) and Fixed Effect (FE), and POLS method allowing the country specific effect to exist indicated it is not the most appropriate model for this study [55, 56]. However, two specification tests namely, Breusch Pagan test and the Hausman test were executed to identify the most suitable model among these three models to determine the regional impact. Furthermore, empirical validation for these models were provided by past literature [33, 34]. Furthermore, scatter plot graphs along with the linear fit of the LPI, GCI and IR were created to analyse the trends of the impact of three independent variables on FDI flow in each country.

## Results

Descriptive statistics, including number of observations, mean value, standard deviation, and minimum and maximum values for FDI, LPI, GCI, and IR, have been presented in S2 Appendix to further examine the dataset. As per the descriptive statistics summary, the Asia and Pacific region is entitled to an average FDI inflow of USD 69.5 billion. The average LPI and GCI values are 4.46 and 3.04 respectively, while recording an average of 4.38.

The results of the Hausman specification test and the Breusch Pagan test RE model were identified as the most suitable technique to analyse the regional impact and the results of the RE model for the Asia and Pacific region. The time series multiple linear regression model employed to analyse the country specific effects of LPI, GCI and IR on FDI are portrayed in S3 Appendix. However, five countries were omitted due to the inability to obtain regression results for the MLR since data was available only for a few years. Findings indicate that even though LPI has no regional impact on FDI, country-specific impact of LPI on FDI flow is seen in several countries. However, multiple studies proved that LPI significantly influences the regional FDI flow [16, 22]. The country-specific impact of LPI on FDI infer that India, Korea, Lebanon and Oman have illustrated significant positive influence on FDI at a 5% significance level. However, in contrast, several countries, including China, and the Philippines as two nations located in East Asia and Kuwait as a Middle East nation, have identified a significant negative impact on investment inflows, where China at a 1% significance level and Kuwait and Philippines at a 10% significant level.

According to the results, the regional impact of GCI on FDI inflow in the Asia & Pacific region has a significant positive impact at a 1% level of significance. However, considering the country-specific effect, GCI for China and Korea indicated a significant positive influence on foreign investment flow at a 1% significant level. Although significant levels were identified at 5%, Kuwait and the Philippines have shown a significant positive impact on FDI. In addition, Pakistan specifies that GCI significantly influences FDI at a 10% significant level. Opposingly, GCI shows a significant negative impact on FDI flow at a 10% significant level in Armenia, India, and Lebanon. However, when closely monitoring the results for LPI and GCI impact, it can be concluded that results of the current study indicate a significant positive influence in consistent with Dunnins' eclectic paradigm theory. Additionally, in some instances, other factors such as a high level of bureaucracy and strict capital control laws, have probably altered the impact of LPI and GCI on FDI.

The RE model results for the Asia & Pacific region indicate IR has a negative impact on regional FDI inflow at a 10% significance level. However, the impact of IR on FDI validated by

past studies indicates a variety of effects where Chandra and Handoyo [48] determined that there is no statistical influence of IR on FDI, while Mishra and Jena [13] suggested the impact is negatively associated. Meanwhile, multiple studies infer that the host country IR has a significant positive relationship with FDI flow [44, 45, 57]. Considering the country-specific impact of IR, the findings emphasise that China and Egypt have a significant coefficient at a 10% significance level with a positive effect on FDI flow, compared to a similar result for Egypt [37]. However, Korea as an East Asian country, indicates a significant negative impact on FDI at a 1% significant level and in Lebanon, IR indicates a significant negative influence on FDI at a 5% level of significance. The mixed result presented by the IR impact can be further justified by the theories of FDI, where the internal fund theory highlights that when the return on investment is high, it significantly influences investment decisions. As per the Keynes theory, the negative impact of IR in certain countries depends on country specific economic deviations. Therefore, the findings align with relevant theories of FDI depending on the country specific characteristics. S4 Appendix portrays the linear fit scatter plot graphs and provides a more prominent analysis of the country-specific trends on how LPI, GCI and IR impact FDI flow.

## Discussion

Considering the findings of the study, it can be suggested that GCI and IR have an impact on the FDI flow in the Asia and Pacific region. Nevertheless, the results of the country-specific analysis point that all three independent variables have a considerable impact on FDI flow in several countries and the nature of the impact differs along with the characteristics of the countries. Possibly, several such characteristics focus on the LPI; even though the regional impact of LPI on FDI was insignificant in the study. A similar study conducted in the Asian region, excluding Middle East and Pacific region countries, identified a significant positive relationship [16]. On top of this, the reason for mixed results with both positive and negative impacts in some countries is that the LPI individually cannot attract FDI. Institutional quality is a key requirement of investors, except for infrastructural quality. This is because if the country is politically unstable, investor-friendly economic conditions would negatively affect investment decisions. Countries with red tape bureaucracy tend to discourage investors concerning the high cost incurred in executing businesses.

In terms of GCI, countries such as China, Korea, Kuwait, Pakistan and the Philippines, have indicated a positive effect, while Armenia, India and Lebanon have indicated a negative effect even though the regional impact of GCI on FDI is positive. As per Alfaki and Ahmed [29], the impact of pillars within GCI could differently influence the impact of GCI on FDI flow in various countries. The authors further highlighted that in the UAE, pillars are related to technological advancement and infer more influence on FDI flow. Moreover, GCI and FDI present a bi-directional influence on each variable, where countries attracting more FDI tend to have the advantage of acquiring cutting-edge technologies and rapid development in all the other areas. Overall, these create an investor-friendly environment.

The impact of IR on FDI flow has shown more inconsistent results in past literature. The findings of the study indicate that even though the regional impact is negative when considering the country specific analysis, countries such as China and Egypt infer a positive influence. Such behaviour explains that investors who expect to inject foreign currency into an economy seek higher IR for a greater return on investment. In contrast, investors who seek local financial sources for business continuity prefer low-interest rates incurring low costs in the business. Therefore, depending on the types of investments in different countries could

significantly impact the nature of IR's influence on the FDI flow. Therefore, this justifies why IR infers drastic changes in its impact on the FDI flow.

## Conclusion

The Asia & Pacific region is the highest FDI attracting destination. Many research studies have been conducted to determine the factors influencing FDI flow in this region. However, the present study sheds light on LPI and GCI as two major indexes evaluate countries' competitiveness and growth potential and IR is a key macroeconomic factor that influences investors' investment behaviour. However, the number of studies conducted to determine how each of these factors influences FDI is comparatively low. Therefore, this study focused on examining the impact of LPI, GCI and IR on FDI in the Asia & Pacific region. The study was conducted for 33 Asia and Pacific counties from 2007 to 2018 including six times series. The analytical technique utilised to investigate the regional impact is the RE model and multiple linear regression was employed to examine the country specific impact on FDI.

The findings infer that GCI has a significant positive influence on FDI inflow which was further backed by past literature [27, 31, 32]. This proves that the GCI index is a sophisticated evaluation tool that investors and policymakers can rely on to make vital decisions on selecting the investment locations, policy implementations as well as planning and initiating regional development initiatives. The results further indicated that IR has a significant negative influence FDI aligning with many past studies [39, 40, 42]. This infers that when the IR rises, it tends to discourage investors from the regional perspective. However, considering the country-specific results, the effect of each variable on FDI flow indicated mixed results with both positive and negative influences based on the selected country's characteristics.

Therefore, concerning recommendations for policy implementation, it can be argued that when countries have better LPI and GCI rankings, its influence on attracting FDI varies based on other factors. Policymakers should not only focus on achieving a high level of macroeconomic and infrastructural development to attract FDI but also ensure other internal factors are favourable(that encourage lowering the cost of conducting business and providing a high return on investments). Furthermore, Asia & Pacific region consists of many developing and emerging economies. Luttermann, Kotzab [22] emphasise that policymakers should prioritise on ensuring high level of logistic performance and competitiveness in order to level up these nations with industrialised developed world to attract more investments. Additionally, maritime countries in the region should pay more attention towards logistics performance enhancement which [58] can significantly influence on attracting FDI. Lastly, policymakers should concentrate more on understanding the bigger picture rather than focusing on a specific set of factors, to achieve higher performance levels in order to sustain the FDI inflow depending on the country specific characters.

Further studies should consider expanding this study's focus, analysing in depth the impact of individual pillars falling under the LPI and GCI index to investigate how various factors influence the investment flow. Similarly, future researchers can focus on examining how different types of IRs impact on attracting FDI and thereby formulating macroeconomic policies favourably.

## Supporting information

**S1 Appendix. Data file.**
(XLSX)

**S2 Appendix. Summary descriptive statistics of variables.**
(DOCX)

**S3 Appendix. RE and MLR model results.**
(DOCX)

**S4 Appendix. Linear fit scatter plot graphs.**
(DOCX)

## Acknowledgments

The authors would like to thank Ms. Gayendri Karunarathne for proof-reading and editing this manuscript.

## Author Contributions

**Conceptualization:** Pasindu Wannisinghe, Ruwan Jayathilaka.

**Data curation:** Pasindu Wannisinghe, Sanjula Jayakody, Sashini Rathnayake, Deshani Wijayasinghe, Naduni Madhavika.

**Formal analysis:** Pasindu Wannisinghe, Sanjula Jayakody, Sashini Rathnayake, Naduni Madhavika.

**Investigation:** Pasindu Wannisinghe, Sanjula Jayakody, Sashini Rathnayake, Naduni Madhavika.

**Methodology:** Pasindu Wannisinghe, Sanjula Jayakody, Sashini Rathnayake, Deshani Wijayasinghe, Ruwan Jayathilaka, Naduni Madhavika.

**Project administration:** Ruwan Jayathilaka, Naduni Madhavika.

**Software:** Pasindu Wannisinghe, Sanjula Jayakody, Sashini Rathnayake.

**Supervision:** Ruwan Jayathilaka, Naduni Madhavika.

**Validation:** Deshani Wijayasinghe, Ruwan Jayathilaka.

**Visualization:** Pasindu Wannisinghe, Sanjula Jayakody, Sashini Rathnayake.

**Writing – original draft:** Pasindu Wannisinghe, Sanjula Jayakody, Sashini Rathnayake, Ruwan Jayathilaka, Naduni Madhavika.

**Writing – review & editing:** Ruwan Jayathilaka.

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
