## [Decision Letter · Decision Letter 0]

1 Nov 2022

PONE-D-22-25222Determining the Influence of LPI, GCI and IR on FDI: A Study on Asia and Pacific RegionPLOS ONE

Dear Dr. Jayathilaka,

Thank you for submitting your manuscript to PLOS ONE. After careful consideration, we feel that it has merit but does not fully meet PLOS ONE’s publication criteria as it currently stands. Therefore, we invite you to submit a revised version of the manuscript that addresses the points raised during the review process.

We look forward to receiving your revised manuscript.

Kind regards,

Wajid Khan

Academic Editor

PLOS ONE

Journal Requirements:

Additional Editor Comments:

Dear Author,

I have reviewed the article entitled “This study examines the impact of the Logistics Performance Index (LPI), Global Competitiveness Index (GCI) and Interest Rates (IR) on Foreign Direct Investment (FDI) for the Asia & Pacific region”. This is an interesting study and the authors have collected a unique dataset for a unique and progressive methodology. However, in my opinion the paper has some shortcomings. Below I have provided numerous remarks on the text:

1. In several instances, I suggest to cite more relevant and recent studies in the introduction and methodology sections.

2. The introduction should be expanded to include importance, uniqueness and contribution to the literature.

3. Please focus on the abstract, in particular some sentences are not clear. You need to revise the abstract.

4. You need state clearly the contributions of the paper. For example, "Consequently, the current paper seeks to make the following contributions to the existing literature. First,, Second,…., Third, …, Fourth,… and so on". The description of the contribution needs to be more forensic, needs to be more focused.

5. The authors should discuss the relevant theories in detail and relate their findings to a specific theory of on Foreign Direct Investment (FDI) in Asia & Pacific region.

6. The inclusion criteria used are not mentioned clearly. The keywords used for publication search are missing.

Reviewers' comments:

Reviewer's Responses to Questions

**Comments to the Author**

1. Is the manuscript technically sound, and do the data support the conclusions?

Reviewer #1: Yes

Reviewer #2: Partly

2. Has the statistical analysis been performed appropriately and rigorously? 

Reviewer #1: Yes

Reviewer #2: Yes

3. Have the authors made all data underlying the findings in their manuscript fully available?

Reviewer #1: Yes

Reviewer #2: Yes

4. Is the manuscript presented in an intelligible fashion and written in standard English?

Reviewer #1: Yes

Reviewer #2: No

5. Review Comments to the Author

Reviewer #1: The paper aims to investigate the effect of LPI, GDI and IR on FDI in selected Asia-Pacific countries over a few discrete years from 2007 to 2018 using panel data econometrics. Apart from a significant level of effort by the authors to make this manuscript a scholastic one with an appropriate structure, a number of serious observations can be pointed out as follows:

Why only LPI, GDI and IR are gathered to examine themselves as the determinants of FDI? A chunk of previous theories and empirics prescribed a number of macroeconomic, open economy macroeconomic, trade and country/region-specific determinants of FDI. LPI, GDI and IR are also included (directly / indirectly) in several past studies to see their relationship with FDI even in case of Asia / ASEAN / Asia-Pacific nations (Saini & Singhania, 2018; Raeskyesa & Suryandaru, 2020; Soh et al., 2021). Even if the present study focuses only on LPI, GDI and IR; a few variables should be controlled in order to see their linkage with FDI through regression analysis.

In connection with the previous point, another important comment is that the theoretical underpinnings of the empirically examined relationship among variables in this study are not well described. Atleast the variable-wise theoretical discussions (with references) about their possible relationships with FDI should be presented.

All of the variables considered here might be suffering from the unit root problems, i.e. the stochastic features in spite of the discrete-time points over a short period. So, authors should perform unit root test(s) in case of both time-series and panel data structures to test whether the variables are stationary at level or not, before conducting the regression analysis.

In the estimated regression models, authors use log values only in the case of FDI as the dependent variable, whereas all regressors are in their actual values. How do they justify it? Do authors use the data on gross FDI or net FDI inflows? Authors should use the net FDI inflow to GDP ratio rather only FDI as the dependent variable.

Authors should not paste any figure directly from its original source. The conclusion of the study should corroborate a few past studies. The basis of selection of countries is not well explained. There are some grammatical and typographical errors that should be rectified.

So, a major revision based on the above-mentioned comments is recommended.

Reviewer #2: Originality: The study lacks the originality to be justified. It lacks generalization aspects of the research study as a number of other variables might be added to enhance the quality of findings. Needs valid justification.

Literature Review: It lacks synthesis to build up the narration and relationship among the variables, under study. Provision of relevant and justified synthetic relationship of the variables in light of empirical literature along with theoretical background is most desirable.

Methodology: In methodology section and subsection of data is lacking proper justification of sub-variables explaining. The methodology is lacking valid literary reasoning. Justification for selection data set; why the six years data of 2007,2010,2012,2014, 2016 and 2018, missing the between years? Why some countries have been excluded within the sample size? Econometric techniques applied, needs further filtration and empirical justification in light of literature.

Variables: Why some key variables like Quality of Governance, Tax treaties and tax system, Balance of Payment, Exchange Rate etc. Provide valid justification in light of literature.

References: Do cite some good empirical papers. Provide valid empirical academic referencing.

Grammar and Language: It is advised to rephrase the text through the paper, improve the language and do a thorough proof read.

Recommendation:

The paper is only accepted if the following major revisions are made and addressed.

Suggested Research Papers:

Some of the relevant papers to be read and cited are as follows:

Shah, M. H. (2014). The significance of infrastructure for FDI inflow in developing countries. Journal of Life Economics, 1(2), 1-16.

Shah, M. H., & Khan, Y. (2016). Trade liberalization and FDI inflows in emerging economies. Shah, MH, & Khan, Y.(2016). Trade Liberalization and FDI Inflows in Emerging Economies. Business & Economic Review, 8(1), 35-52.

Shah, M. H., & Afridi, A. G. (2015). Significance of good governance for FDI inflows in SAARC countries. Shah, MH, & Afridi, AG (2015). Significance of Good Governance for FDI Inflows in SAARC Countries. Business & Economic Review, 7(2), 31-52.

Shah, M. H. (2017). The effect of macroeconomic stability on inward FDI in African developing countries. Shah, MH,(2016). The effect of macroeconomic stability on inward FDI in African developing countries. International Journal of Business Studies Review, 1(1), 1-11.

Ullah, Z., Shah, M. H., Khan, W., & Ali, A. (2021). Macroeconomic Factors As Drivers Of Foreign Portfolio Investment In Emerging Economy. Multicultural Education, 7(6).

Etc.

6. PLOS authors have the option to publish the peer review history of their article (what does this mean?). If published, this will include your full peer review and any attached files.

Reviewer #1: No

Reviewer #2: No

---

## [Author Response · Author response to Decision Letter 0]

14 Dec 2022

Point by point response to reviewers

Dear editor and reviewers.

Thank you for giving us the opportunity to submit a revised draft of the manuscript “Determining the Influence of LPI, GCI and IR on FDI: A Study on Asia and Pacific Region” for publication in the prestigious “PLOS ONE” journal. We appreciate the time and effort that you have dedicated to providing feedback on our manuscript and are grateful for the insightful comments on and valuable improvements to our paper. We have incorporated most of the suggestions made by the editor and respective reviewers. Those changes are highlighted within the manuscript. Please see below, for a point-by-point response to the reviewers’ comments and concerns. All page numbers refer to the revised manuscript file with tracked changes.

Editor Comments:

Editor’s general commnet I have reviewed the article entitled “This study examines the impact of the Logistics Performance Index (LPI), Global Competitiveness Index (GCI) and Interest Rates (IR) on Foreign Direct Investment (FDI) for the Asia & Pacific region”. This is an interesting study and the authors have collected a unique dataset for a unique and progressive methodology. However, in my opinion the paper has some shortcomings.

Authors’ Response: Well noted your comment. The shortcomings pointed out are improved as suggested, and the below responses and actions taken for the highlighted shortcoming are given.

Editors Comment 1: In several instances, I suggest citing more relevant and recent studies in the introduction and methodology sections.

Authors’ Response: Thank you for the comment. The introduction and methodology sections have been updated, having cited recent and relevant studies as suggested. Such that, 

“ In general, FDI is a form of cross border investment where investors residing in one country invest in another country/economy and can significantly influence investee entity’s economic activities entity beyond the border [2]. In recent years, global FDI has become major phenomenon owing to economic interconnectivity, globalisation, the rapid growth of international trade, inter-regional and intra-regional trade agreements, among others [3]. As per the UNCTAD [4], developing countries have attracted a significant portion of the FDI inflow. Likewise, Buthe and Milner [5] specified that…” Refer lines 48 to 57.

“ On the other hand, logistic capabilities and transportation have become major determinants of attracting FDI and achieving sustainable economic growth [10]” Refer lines 80 to 81.

“ Binsbergen (2022) stresses that IR is a recognised tool to calculate the time value of money or to determine the investment yield from a futuristic perspective.” Refer lines 91 to 93.

“ As per Jayathilaka, Jayawardhana [54], the limited number of time series is a constraint in utilising non-stationary and dynamic panel data models.” Refer lines 273 to 275.

“… POLS method allowing the country specific effect to exist indicated it is not the most appropriate model for this study [55, 56]. ” Refer lines 277 to 278.

“ Furthermore, empirical validation for these models were provided by past literature[33, 34].” Refer lines 281 to 282.

Editors Comment 2: The introduction should be expanded to include importance, uniqueness and contribution to the literature.

Authors’ Response: Noted with thanks. We have added the suggested content to the manuscript on importance, uniqueness and contribution to the literature. 

“ Hence, LPI and GCI are justified as two globally recognised benchmarking tools. Here, LPI provides comprehensive insights on trade logistic performance and competitiveness of countries for governments and global enterprises to strategise their investment decisions; IR is a key component of a country’s monetary policy that impact investment decisions. Therefore, it raises the question of how these factors individually and collectively influence the FDI flow in the Asia and Pacific region. To further investigate the impact, this study objectified determining the impact of LPI, GCI and IR on FDI flow in the Asia and Pacific region. ” Refer lines 104 to 112.

“ Consequently, the current paper seeks to contribute the following to the existing literature. Firstly,…..” (Refer Editor Comment 4). Refer lines 113 to 114.

Editors Comment 3: Please focus on the abstract, in particular some sentences are not clear. You need to revise the abstract.

Authors’ Response: Noted with thanks. This has been incorporated in the revised manuscript.

“The study is original as extensive evidence on the impact of LPI, GCI and IR on FDI in the Asia & Pacific region are examined initially” Refer lines 28 to 29.

Editors Comment 4: You need state clearly the contributions of the paper. For example, "Consequently, the current paper seeks to make the following contributions to the existing literature. First, Second,…., Third, …, Fourth,… and so on". The description of the contribution needs to be more forensic, needs to be more focused.

Authors’ Response: Thank you for pointing out this out. As you suggested the incorporated change with contribution of the paper is below.

“Consequently, the current paper seeks to contribute the following to the existing literature. Firstly, the study analyses the overall impact of LPI, GCI and IR on FDI, focusing on Asia and Pacific region countries, where a lacuna in the literature was observed. Secondly, the study provides a country specific analysis focusing on Asia and Pacific countries in determining how these three independent variables influence the investment flow within countries- i.e. how these three major factors determine the quality and the productivity of macroeconomic and microeconomic conditions within countries. Thirdly, the study focuses on understanding the differences in the linear trends of how all three variables impact countries individually, utilising graphical illustrations to visualise the trends. ” Refer lines 113 to 122.

Editors Comment 5: The authors should discuss the relevant theories in detail and relate their findings to a specific theory of on Foreign Direct Investment (FDI) in Asia & Pacific region.

Authors’ Response: Well noted and thank you. Having accepted the suggestion, the relevant theories of FDI were identified and further discussed in both the literature review and findings sections.

“In addition to that, the author disclosed that when considering the theoretical perspective of FDI, theories such as Dunning’s eclectic paradigm theory which emphasise the fluctuations of FDI inflow with relation to multiple factors including efficiency which significantly rely on competitiveness, infrastructural capabilities, logistic performance and quality of governance is key theory of FDI aligning with study.” Refer lines 148 to 153.

“Furthermore, multiple theories related to cross border investment aligning with macro-economic factors have also been covered under the scope of this study. Suhendra, Istikomah [50] emphasised the internal fund theory, according to which, the investment decision is based on the level of returns considering multiple factors and the return on investment including IR. On the other hand, Ajija and Fanani [51] pointed out that the Keynes theory infers that when the IR increases, the FDI inflow reduces while indicating a negative impact.” Refer lines 227 to 233.

“However, when closely monitoring the results for LPI and GCI impact, it can be concluded that results of the current study indicate a significant positive influence in consistent with Dunnins' eclectic paradigm theory. Additionally, in some instances, other factors such as a high level of bureaucracy and strict capital control laws, have probably altered the impact of LPI and GCI on FDI. ” Refer lines 316 to 321.

“The mixed result presented by the IR impact can be further justified by the theories of FDI, where the internal fund theory highlights that when the return on investment is high, it significantly influences investment decisions. As per the Keynes theory, the negative impact of IR in certain countries depends on country specific economic deviations. Therefore, the findings align with relevant theories of FDI depending on the country specific characteristics.” Refer lines 333 to 339.

Editors Comment 6: The inclusion criteria used are not mentioned clearly. The keywords used for publication search are missing

Authors’ Response: Thank you. The comment is well received. Reviewing the past literature it was noted that, foreign direct investments, logistic performance index, global competitiveness index, interest rates were used numerously as keywords (Soh et al., 2021) (Luttermann et al., 2020) (Mishra and Jena, 2019) (Alfaki and Ahmed, 2013). Therefore, the current study sticks to the stated keywords having used Foreign direct investments, logistic performance index, global competitiveness index, interest rates as the keywords.

Reviewer 1 General Comment: The paper aims to investigate the effect of LPI, GDI and IR on FDI in selected Asia-Pacific countries over a few discrete years from 2007 to 2018 using panel data econometrics. Apart from a significant level of effort by the authors to make this manuscript a scholastic one with an appropriate structure, a number of serious observations can be pointed out as follows:

Authors’ Response: Well noted and thank you. Based on the improvements pointed out, the manuscript was revised as below.

Reviewer 1 Comment 1: Why only LPI, GDI and IR are gathered to examine themselves as the determinants of FDI? 

A chunk of previous theories and empirics prescribed a number of macroeconomic, open economy macroeconomic, trade and country/region-specific determinants of FDI. LPI, GDI and IR are also included (directly / indirectly) in several past studies to see their relationship with FDI even in case of Asia / ASEAN / Asia-Pacific nations (Saini & Singhania, 2018; Raeskyesa & Suryandaru, 2020; Soh et al., 2021). 

Even if the present study focuses only on LPI, GDI and IR; a few variables should be controlled in order to see their linkage with FDI through regression analysis.

Authors’ Response: Well noted and thank you. Based on the improvements pointed out, the manuscript was revised as below.

Thank you for raising this question. The importance of investigating the LPI, GCI and IR are further strengthened in the paper to further justify the importance of this study. 

The reason to focus on LPI, GCI and IR are because LPI is global benchmarking tool that provides comprehensive knowledge on the trade logistic performance of countries to identify their challenges and opportunities as well the means to improve the performance levels. Therefore, it has become a vital tool for investors to identify investment locations. Hence this study focuses on understanding how the variations in LPI effected on the FDI inflow. 

When it comes to GCI, it is holistic index that cover almost every aspect of macro and microeconomic factor that influence the competitiveness of countries under 12 major pillars. Therefore, it provides crucial insights to governments and global businesses to understand the competitiveness of countries when engaging in economic activities. As a result, this study stresses on importance of identifying how the variations in GCI ranking have impacted on FDI inflow in the Asia & Pacific region. 

On the other hand, the reason to consider IR is because it is key element in country’s monetary policy which ultimately impact on the investment behavior of investors. Therefore, it was considered as an independent variable in this study to further investigate effect of IR fluctuations on FDI inflow. 

“Hence, LPI and GCI are justified as two globally recognised benchmarking tools. Here, LPI provides comprehensive insights on trade logistic performance and competitiveness of countries for governments and global enterprises to strategise their investment decisions; IR is a key component of a country’s monetary policy that impact investment decisions.” Refer lines 104 to 108.

We agree that this is a potential limitation of the study. However, the reason to omit the other determinants highlighted in past literature is because LPI and GCI indexes fully or partially consider most of these determinants within their evaluation criteria. To further elaborate, LPI methodology consider 6 different sub pillar which are Efficiency of customs and border clearance, Quality of trade and transport infrastructure, Ease of arranging competitively priced shipments, Competence and quality of logistics services, Ability to track and trace consignment and the Frequency with which shipments reach consignees within scheduled or expected delivery times.

Furthermore, GCI index is created utilizing 12 major pillars considering 98 sub pillars. 12 major pillars include, Institutions, Infrastructure, ICT adoption, Macroeconomic stability, Health, Skills, Product market, Labour market, Financial system, Market size, Business dynamism and the Innovation capability.

Reviewer 1 Comment 2: In connection with the previous point, another important comment is that the theoretical underpinnings of the empirically examined relationship among variables in this study are not well described. Atleast the variable-wise theoretical discussions (with references) about their possible relationships with FDI should be presented.

Authors’ Response: Well noted the comment with thanks. Accordingly, we have further empirically explained the possible relationships of the variables with FDI with related references. 

“Moreover, LPI is considered a factor to positively impact FDI inflows into a country since adequate infrastructure facilities, efficient transportation systems, etc., boost a country's logistics performance. Consequently, trade and investments will thrive since better logistics performance facilitates trade between countries. Also, investors prefer to invest in countries with better infrastructure and logistics facilities, considering these lucrative.” Refer lines 137 to 144.

“Similar to LPI, GCI too positively impacts FDI inflows since factors such as institutional quality, infrastructure, macroeconomic stability, skills, innovation etc., of a country tend to encourage investors to invest in such countries[27]. ” Refer lines 175 to 177.

“The impact of IR on FDI depends on the type of IR under consideration. Borrowing IR negatively impacts FDI inflows since investors tend to invest in countries with low borrowing IR to lower the cost of capital. In contrast, lending IR positively impacts FDI since investors tend to invest in countries with higher IR to ensure a higher return on their investments [35]. ” Refer lines 193 to 200.

Reviewer 1 Comment 3: All of the variables considered here might be suffering from the unit root problems, i.e. the stochastic features in spite of the discrete-time points over a short period. So, authors should perform unit root test(s) in case of both time-series and panel data structures to test whether the variables are stationary at level or not, before conducting the regression analysis.

Authors’ Response: Thank you. The comment is well received. Unit root test or testing stationarity is simply focused on identifying whether the statistical properties of data changes over the time. However, the reason for not conducting the unit root test in this study is because, in the current study even though the panel dataset consists of 33 cross sections it contains only 6 time series which are 2007, 2010, 2012, 2014, 2016, and 2018 with uneven gaps. Therefore, the dataset is not only unbalanced but also consist of uneven gaps in the time series. Therefore, it was not possible to conduct the unit root test in this study.

Reviewer 1 Comment 4: In the estimated regression models, authors use log values only in the case of FDI as the dependent variable, whereas all regressors are in their actual values. How do they justify it? Do authors use the data on gross FDI or net FDI inflows? Authors should use the net FDI inflow to GDP ratio rather only FDI as the dependent variable.

Authors’ Response: Thank you for the comment and it is well noted. Initially, the natural log values were taken to ensure the uniformity of the dataset.

However, considering the comment we have conducted the analysis using the net FDI inflow as percentage of GDP to further strengthen the validity of the results. 

However, the results for the Panel model were insignificant with higher P values and the country specific analysis indicated few countries with significant results. Therefore, to improve the quality of the coverage of the results it was decided to continue the study with net FDI inflows. Nevertheless, we have included the results obtained using the net FDI inflow to GDP ratio in the appendices for your reference. 

To further justify, there are few past studies which have utilized the net FDI inflow to determine the determinants of FDI which are (Hossain and Ahmed, 2018), (Cruz and Siy, 2018), (Mengistu and Adhikary, 2011), (Kumari and Sharma, 2017).

Reviewer 1 Comment 5: Authors should not paste any figure directly from its original source. The conclusion of the study should corroborate a few past studies. 

The basis of selection of countries is not well explained. 

There are some grammatical and typographical errors that should be rectified.

Authors’ Response: Appreciate the comment. It was incorporated with having removed the cited figure. Further, conclusion was validated by including multiple references for results and recommendations. 

“.. which was further backed by past literature [27, 31, 32].” Refer lines 394 to 395.

“ FDI aligning with many past studies [39, 40, 42]. ” Refer lines 399 .

“Furthermore, Asia & Pacific region consists of many developing and emerging economies. Luttermann, Kotzab [22] emphasise that policymakers should prioritise on ensuring high level of logistic performance and competitiveness in order to level up these nations with industrialised developed world to attract more investments. Additionally, maritime countries in the region should pay more attention towards logistics performance enhancement which [58] can significantly influence on attracting FDI. Lastly, policymakers should concentrate more on understanding the bigger picture rather than focusing on a specific set of factors, to achieve higher performance levels in order to sustain the FDI inflow depending on the country specific characters.” Refer lines 409 to 418.

The selection of countries was limited to 33 countries, and it was selected based on the availability of the secondary data. The methodology section is further improved to highlight the selection process.

“The approach utilised to determine the number of countries strongly relied on the availability of secondary data for variables.” Refer lines 248 to 249.

The grammatical and typographical errors were eliminated after thorough proofread of the manuscript.

Reviewer 2 Comment 1: Originality: The study lacks the originality to be justified. It lacks generalization aspects of the research study as a number of other variables might be added to enhance the quality of findings. Needs valid justification.

Authors’ Response: Well noted and thank you for the comment. There are other variables that affects FDI. However, the reason those variables have not been directly considered in the current study is because the LPI and GCI indexes fully or partially cover a wide range of variables within its sub pillars and it was our intention not to repeat the similar variables. We have provided an empirical justification on these variables as illustrated. Further, kindly refer Reviewer #1 Comment 1.

“ Here, LPI includes 6 sub pillars such as customs, infrastructure, international shipments, quality of logistics services, tracking and tracing and timeliness, while GCI includes 12 major pillars, such as institutions, infrastructure, ICT adoption, macroeconomic stability, health, skills, product market, labour market, financial system, market size, business dynamism and innovation capability. Under these 12 pillars, there are another 98 sub pillars [52, 53].” Refer lines 254 to 259.

Reviewer 2 Comment 2: Literature Review: It lacks synthesis to build up the narration and relationship among the variables, under study. Provision of relevant and justified synthetic relationship of the variables in light of empirical literature along with theoretical background is most desirable.

Authors’ Response: Thank you and well noted on the comment. The literature review was further revised by providing relevant and justified synthesized relationship of the variables in light of empirical literature. 

“ Similarly, An, Razzaq [17] and Saidi, Mani [10] had concluded that logistics performance is a determinant of FDI” Refer line 153 to154.

“ Affirming the above-explained findings, Avioutskii and Tensaout [19] examined the importance of logistics infrastructure in attracting FDI into the European region in a similar study..” Refer line 156 to159.

“ Furthermore, Shah [25] had identified that infrastructure availability in the host country has a positive impact on the location choices of foreign investors. Further, the author determined that macroeconomic management and economic development positively impact FDI, while high inflation affects FDI negatively. Also, according to Shah [26] better infrastructure and trade liberalisation encourage more FDI inflows into a country.” Refer line 168 to173.

“ Similar to LPI, GCI too positively impacts FDI inflows since factors such as institutional quality, infrastructure, macroeconomic stability, skills, innovation etc., of a country tend to encourage investors to invest in such countries[27].” Refer line 175 to177.

“ Moreover, GCI includes many other variables under its sub pillars. When considering the impact of those variables on FDI, trade and investment liberalisation, market size, development level, human capital, political stability, regulatory quality, the openness of the host economy and good governance have a positive impact on FDI while corruption negatively impacts FDI [33, 34].” Refer line 186 to 190.

“ The impact of IR on FDI depends on the type of IR under consideration. Borrowing IR negatively impacts FDI inflows since investors tend to invest in countries with low borrowing IR to lower the cost of capital. In contrast, lending IR positively impacts FDI since investors tend to invest in countries with higher IR to ensure a higher return on their investments [35].” Refer line 193 to 200.

“ In addition, economic development and IR risk positively impact Foreign Portfolio Investments (FPI), while inflation, exchange rates and country risk have a negative impact on FPI [49].” Refer line 222 to 224.

“… In addition to that, the author disclosed that when considering the theoretical perspective of FDI, theories such as Dunning’s eclectic paradigm theory which emphasise the fluctuations of FDI inflow with relation to multiple factors including efficiency which significantly rely on competitiveness, infrastructural capabilities, logistic performance and quality of governance is key theory of FDI aligning with study.” Refer lines 148 to 153.

 “ Furthermore, multiple theories related to cross border investment aligning with macro-economic factors have also been covered under the scope of this study. Suhendra, Istikomah [50] emphasised the internal fund theory, according to which, the investment decision is based on the level of returns considering multiple factors and the return on investment including IR. On the other hand, Ajija and Fanani [51] pointed out that the Keynes theory infers that when the IR increases, the FDI inflow reduces while indicating a negative impact.” Refer lines 227 to 233.

Reviewer 2 Comment 3: Methodology: In methodology section and subsection of data is lacking proper justification of sub-variables explaining. 

The methodology is lacking valid literary reasoning. Justification for selection data set; why the six years data of 2007,2010,2012,2014, 2016 and 2018, missing the between years? 

Why some countries have been excluded within the sample size? 

Econometric techniques applied, needs further filtration and empirical justification in light of literature.

Authors’ Response: Thank you and well noted on the comment. Sub variables included in the LPI and GCI were further explained and provided a holistic understanding on the manuscript from lines 254 to 259 . Refer Reviewer #2 Comment 1.

The reason to limit the study for 2007,2010,2012,2014, 2016 and 2018 is that the data for the LPI is published only for those 6 time periods and the year consist uneven gaps as well. Revisions were made in the manuscript to further explaining this. Refer lines 224 to 246.

Furthermore, the sample size was limited to 33 countries due to the availability of secondary data where IR has the lowest number of country inclusion that has resulted in limiting the number of countries to 33. However, it is further explained in the manuscript in refer lines 248 to 249

“The approach utilised to determine the number of countries strongly relied on the availability of secondary data for variables.” 

Further justification was provided to the selection of the econometric technique with the light of the literature. 

“As per Jayathilaka, Jayawardhana [54], the limited number of time series is a constraint in utilising non-stationary and dynamic panel data models.” Refer lines 273 to 275.

“…POLS method allowing the country specific effect to exist indicated it is not the most appropriate model for this study [55, 56].” Refer lines 277 to 278.

“Furthermore, empirical validation for these models were provided by past literature[33, 34].” Refer lines 281 to 282.

Reviewer 2 Comment 4: Variables: Why some key variables like Quality of Governance, Tax treaties and tax system, Balance of Payment, Exchange Rate etc. Provide valid justification in light of literature.

Authors’ Response: Thank you and well noted on the comment. As elaborated in the Reviewer #1 Comment 1 and Reviewer #2 Comment 1, these variables were not separately evaluated since the 6 sub pillars in LPI and the 12 major pillars with 98 sub pillars of GCI cover these variables. The impact of these variables are evaluated through the LPI and GCI. We have provided further justification on this in the lines 254 to 259.

Reviewer 2 Comment 5: References: Do cite some good empirical papers. Provide valid empirical academic referencing.

Authors’ Response: Thank you and well noted on the comment. We have cited more empirical papers with empirical academic referencing.

“ In recent years, global FDI has become major phenomenon owing to economic interconnectivity, globalisation, the rapid growth of international trade, inter-regional and intra-regional trade agreements, among others [3]. ” Refer line 51 to 54.

“Similarly, An, Razzaq [17] and Saidi, Mani [10] concluded that logistics performance is a determinant of FDI.” Refer line 153 to 154.

“Affirming the above-explained findings, Avioutskii and Tensaout [19] examined the importance of logistics infrastructure in attracting FDI into the European region in a similar study.” Refer line 156 to 159.

“Furthermore, Shah [25] had identified that infrastructure availability in the host country has a positive impact on the location choices of foreign investors. Further, the author determined that macroeconomic management and economic development positively impact FDI, while high inflation affects FDI negatively. Also, according to Shah [26] better infrastructure and trade liberalisation encourage more FDI inflows into a country.” Refer line 168 to 173.

“Similar to LPI, GCI too positively impacts FDI inflows since factors such as institutional quality, infrastructure, macroeconomic stability, skills, innovation etc., of a country tend to encourage investors to invest in such countries[27].” Refer line 175 to 177.

“Moreover, GCI includes many other variables under its sub pillars. When considering the impact of those variables on FDI, trade and investment liberalisation, market size, development level, human capital, political stability, regulatory quality, the openness of the host economy and good governance have a positive impact on FDI while corruption negatively impacts FDI [33, 34].” Refer line 186 to 190.

“The impact of IR on FDI depends on the type of IR under consideration. Borrowing IR negatively impacts FDI inflows since investors tend to invest in countries with low borrowing IR to lower the cost of capital. In contrast, lending IR positively impacts FDI since investors tend to invest in countries with higher IR to ensure a higher return on their investments [35].” Refer line 193 to 200.

“In addition, economic development and IR risk positively impact Foreign Portfolio Investments (FPI), while inflation, exchange rates and country risk have a negative impact on FPI [49].” Refer line 222 to 224.

“In addition to that, the author disclosed that when considering the theoretical perspective of FDI, theories such as Dunning’s eclectic paradigm theory which emphasise the fluctuations of FDI inflow with relation to multiple factors including efficiency which significantly rely on competitiveness, infrastructural capabilities, logistic performance and quality of governance is key theory of FDI aligning with study.”. Refer line 148 to 153.

“Furthermore, multiple theories related to cross border investment aligning with macro-economic factors have also been covered under the scope of this study. Suhendra, Istikomah [50] emphasised the internal fund theory, according to which, the investment decision is based on the level of returns considering multiple factors and the return on investment including IR. On the other hand, Ajija and Fanani [51] pointed out that the Keynes theory infers that when the IR increases, the FDI inflow reduces while indicating a negative impact. ” Refer line 227 to 233.

“ As per Jayathilaka, Jayawardhana [54], the limited number of time series is a constraint in utilising non-stationary and dynamic panel data models.”Refer line 273 to 275.

Reviewer 2 Comment 6: Grammar and Language: It is advised to rephrase the text through the paper, improve the language and do a thorough proofread.

Suggested Research Papers:

Some of the relevant papers to be read and cited are as follows:

Shah, M. H. (2014). The significance of infrastructure for FDI inflow in developing countries. Journal of Life Economics, 1(2), 1-16.

Shah, M. H., & Khan, Y. (2016). Trade liberalization and FDI inflows in emerging economies. Shah, MH, & Khan, Y.(2016). Trade Liberalization and FDI Inflows in Emerging Economies. Business & Economic Review, 8(1), 35-52.

Shah, M. H., & Afridi, A. G. (2015). Significance of good governance for FDI inflows in SAARC countries. Shah, MH, & Afridi, AG (2015). Significance of Good Governance for FDI Inflows in SAARC Countries. Business & Economic Review, 7(2), 31-52.

Shah, M. H. (2017). The effect of macroeconomic stability on inward FDI in African developing countries. Shah, MH,(2016). The effect of macroeconomic stability on inward FDI in African developing countries. International Journal of Business Studies Review, 1(1), 1-11.

Ullah, Z., Shah, M. H., Khan, W., & Ali, A. (2021). Macroeconomic Factors As Drivers Of Foreign Portfolio Investment In Emerging Economy. Multicultural Education, 7(6).

Etc.

Authors’ Response: Noted with thank you. This has been incorporated in the revised manuscript.

Noted with thank you, for the interesting studies that have been provided and we have included of these studies as well.

“ Furthermore, Shah [25] had identified that infrastructure availability in the host country has a positive impact on the location choices of foreign investors. Further, the author determined that macroeconomic management and economic development positively impact FDI, while high inflation affects FDI negatively.” Refer lines 168 to 173.

“….. Moreover, GCI includes many other variables under its sub pillars. When considering the impact of those variables on FDI, trade and investment liberalisation, market size, development level, human capital, political stability, regulatory quality, the openness of the host economy and good governance have a positive impact on FDI while corruption negatively impacts FDI [33, 34].”Refer lines 186 to 190.

“…… Also, according to Shah [26] better infrastructure and trade liberalisation encourage more FDI inflows into a country. ” Refer lines 172 to 173.

“…. In addition, economic development and IR risk positively impact Foreign Portfolio Investments (FPI), while inflation, exchange rates and country risk have a negative impact on FPI [49]. ” Refer lines 222 to 224.

Additional Reviewer 2 Comments

Additional Reviewer 2 Comment 1: Improve originality of the study by citing latest, most relevant and high impact factor research papers in the field.

Authors’ Response: Duly noted and thank you. Latest and high impact factor research paper were cited in order to improve he originality of the study. Kindly refer Reviewer #02 Comment 05.

“In recent years, global FDI has become major phenomenaon as a result ofowing to the economic interconnectivity, globalizsation, the rapid growth of international trade, inter-regional and intra-regional trade agreements, and many more reasonsamong others [3].” Refer line 51 to 54.

“Similarly, An, Razzaq [17] and Saidi, Mani [10] concluded that logistics performance is a determinant of FDI.” Refer line 153 to 154.

“Affirming the above-explained findings, Avioutskii and Tensaout [19] examined the importance of logistics infrastructure in attracting FDI into the European region in a similar study.” Refer line 156 to 159.

“Furthermore, Shah [25] had identified that infrastructure availability in the host country has a positive impact on the location choices of foreign investors. Further, the author determined that macroeconomic management and economic development positively impact FDI, while high inflation affects FDI negatively. Also, according to Shah [26] better infrastructure and trade liberalisation encourage more FDI inflows into a country.” Refer line 168 to 173.

“Similar to LPI, GCI too positively impacts FDI inflows since factors such as institutional quality, infrastructure, macroeconomic stability, skills, innovation etc., of a country tend to encourage investors to invest in such countries[27].” Refer line 175 to 177.

“Moreover, GCI includes many other variables under its sub pillars. When considering the impact of those variables on FDI, trade and investment liberalisation, market size, development level, human capital, political stability, regulatory quality, the openness of the host economy and good governance have a positive impact on FDI while corruption negatively impacts FDI [33, 34].” Refer line 186 to 190.

“The impact of IR on FDI depends on the type of IR under consideration. Borrowing IR negatively impacts FDI inflows since investors tend to invest in countries with low borrowing IR to lower the cost of capital. In contrast, lending IR positively impacts FDI since investors tend to invest in countries with higher IR to ensure a higher return on their investments [35].” Refer line 193 to 200.

“In addition, economic development and IR risk positively impact Foreign Portfolio Investments (FPI), while inflation, exchange rates and country risk have a negative impact on FPI [49].” Refer line 222 to 224.

“In addition to that, the author disclosed that when considering the theoretical perspective of FDI, theories such as Dunning’s eclectic paradigm theory which emphasise the fluctuations of FDI inflow with relation to multiple factors including efficiency which significantly rely on competitiveness, infrastructural capabilities, logistic performance and quality of governance is key theory of FDI aligning with study.”. Refer line 148 to 153.

“Furthermore, multiple theories related to cross border investment aligning with macro-economic factors have also been covered under the scope of this study. Suhendra, Istikomah [50] emphasised the internal fund theory, according to which, the investment decision is based on the level of returns considering multiple factors and the return on investment including IR. On the other hand, Ajija and Fanani [51] pointed out that the Keynes theory infers that when the IR increases, the FDI inflow reduces while indicating a negative impact. ” Refer line 227 to 233

“ As per Jayathilaka, Jayawardhana [54], the limited number of time series is a constraint in utilising non-stationary and dynamic panel data models.” Refer line 273 to 275

Additional Reviewer 2 Comment 2: The study has investigated low quality data as the author has mentioned himself; it creates the problem of originality. It is recommended to improve the data and data variables.

Authors’ Response: Thank you and the comment is well received. Our intention was not to undermine the datasets that were obtained from World Bank – World Development Indicators and World Economic Forum - Global Competitiveness Report which are two most reliable sources of global secondary data as per many past literature. The only limitation is that the LPI index is published only for 6 time series, but the validity and the quality of the dataset can be further justified by looking at the past studies such as (Soh, Wong and Tang, 2021) and (Jayathilaka et al., 2022) since these studies have employed LPI as a major variable.

Additional Reviewer 2 Comment 3: Literature review needs proper synthesis to explain the research gap, research question, hypothesis, variables selection and their relationship and the adoption of empirical techniques.

Authors’ Response: Thank you and well noted on the comment. The literature review was further revised by empirically elaborating the research gap and research question.

“Moreover, LPI is considered a factor to positively impact FDI inflows into a country since adequate infrastructure facilities, efficient transportation systems, etc., boost a country's logistics performance. Consequently, trade and investments will thrive since better logistics performance facilitates trade between countries. Also, investors prefer to invest in countries with better infrastructure and logistics facilities, considering these lucrative.” Refer lines 137 to 144.

“Similar to LPI, GCI too positively impacts FDI inflows since factors such as institutional quality, infrastructure, macroeconomic stability, skills, innovation etc., of a country tend to encourage investors to invest in such countries[27].” Refer line 175 to 177.

“The impact of IR on FDI depends on the type of IR under consideration. Borrowing IR negatively impacts FDI inflows since investors tend to invest in countries with low borrowing IR to lower the cost of capital. In contrast, lending IR positively impacts FDI since investors tend to invest in countries with higher IR to ensure a higher return on their investments [35].” Refer line 193 to 200.

“Past literature stresses that limited studies have analysed the individual impact of LPI, GCI and IR on FDI in the Asia & Pacific region. Moreover, according to our knowledge, no studies were conducted to analyse the overall impact of all three variables on FDI inflow. Therefore, the current study is dedicated to filling the above-mentioned research gap in the existing literature through a comprehensive analysis by determining the impact of LPI, GCI and IR on FDI in the Asia & Pacific region. ” Refer lines 234 to 240.

Additional Reviewer 2 Comment 4: The research design do not seems appropriate to address the research question. Its suggested that either adopt a better research design or convey the research question more clearly.

Authors’ Response: Well noted and thank you for the comment. The manuscript is revised to highlight the importance of variables utilized in this study followed by the research question and the objective of the study from lines 108 to 112.

“Therefore, it raises the question of how these factors individually and collectively influence the FDI flow in the Asia and Pacific region. To further investigate the impact, this study objectified determining the impact of LPI, GCI and IR on FDI flow in the Asia and Pacific region.”

Additional Reviewer 2 Comment 5: Data variables need proper justification of selections and how they address the research problem?

Authors’ Response: Thank you and well noted on the comment. There are many factors influencing FDI inflows. However, , we have selected LPI, GCI and IR since the LPI and GCI covers most of the factors affecting FDI through their sub pillars and the overall impact of these variables could be identified. Further, IR is a major factor in influencing investment decisions. Therefore, it was decided that these variables are most suitable for our study and addressing the research problem since there is lack of studies conducted to identify the impact of these factors on FDI inflows. Kindly refer response for Reviewer#1 Comment 02 for further explanation. Furthermore, lines numbers provide a clearer understanding of the variables.

“Hence, LPI and GCI are justified as two globally recognised benchmarking tools. Here, LPI provides comprehensive insights on trade logistic performance and competitiveness of countries for governments and global enterprises to strategise their investment decisions; IR is a key component of a country’s monetary policy that impact investment decisions.” Refer lines 104 to 108.

“Moreover, LPI is considered a factor to positively impact FDI inflows into a country since adequate infrastructure facilities, efficient transportation systems, etc., boost a country's logistics performance. Consequently, trade and investments will thrive since better logistics performance facilitates trade between countries. Also, investors prefer to invest in countries with better infrastructure and logistics facilities, considering these lucrative.” Refer lines 137 to 144.

“Similar to LPI, GCI too positively impacts FDI inflows since factors such as institutional quality, infrastructure, macroeconomic stability, skills, innovation etc., of a country tend to encourage investors to invest in such countries[27].”Refer lines 175 to 177.

“The impact of IR on FDI depends on the type of IR under consideration. Borrowing IR negatively impacts FDI inflows since investors tend to invest in countries with low borrowing IR to lower the cost of capital. In contrast, lending IR positively impacts FDI since investors tend to invest in countries with higher IR to ensure a higher return on their investments [35].”Refer lines 193 to 200.

Additional Reviewer 2 Comment 6: The language needs a lot of improvement (grammatically and synthetically). A thorough proofread is recommended.

Authors’ Response: Noted and thank you. This has been

incorporated in the revised manuscript.

Additional Reviewer 2 Comment 7: The recommendations given are too general, but the study claims itself to be specific. It needs further elaboration. Supportive justification is need for the findings and conclusion in light of some good research studies.

Authors’ Response: Thank you for the comment and it is well incorporated in the revised paper. The recommendations were further elaborated and justified using the light of the past literature provide more vital and valid recommendations. The changes are included from lines 409 to 418.

“..Furthermore, Asia & Pacific region consists of many developing and emerging economies. Luttermann, Kotzab [22] emphasise that policymakers should prioritise on ensuring high level of logistic performance and competitiveness in order to level up these nations with industrialised developed world to attract more investments. Additionally, maritime countries in the region should pay more attention towards logistics performance enhancement which [58] can significantly influence on attracting FDI. Lastly, policymakers should concentrate more on understanding the bigger picture rather than focusing on a specific set of factors, to achieve higher performance levels in order to sustain the FDI inflow depending on the country specific characters.”

---

## [Decision Letter · Decision Letter 1]

19 Jan 2023

Determining the Influence of LPI, GCI and IR on FDI: A Study on Asia and Pacific Region

PONE-D-22-25222R1

Dear Dr. Ruwan,

We’re pleased to inform you that your manuscript has been judged scientifically suitable for publication and will be formally accepted for publication once it meets all outstanding technical requirements.

Kind regards,

Wajid Khan

Academic Editor

PLOS ONE

---

## [Editor Report · Acceptance letter]

23 Jan 2023

PONE-D-22-25222R1 

Determining the Influence of LPI, GCI and IR on FDI: Study on the Asia and Pacific Region 

Dear Dr. Jayathilaka:

I'm pleased to inform you that your manuscript has been deemed suitable for publication in PLOS ONE. Congratulations! Your manuscript is now with our production department. 

Kind regards, 

on behalf of

Dr. Wajid Khan 

Academic Editor

PLOS ONE